# An Insight on Functioning Pancreatic Neuroendocrine Neoplasms

**DOI:** 10.3390/biomedicines11020303

**Published:** 2023-01-21

**Authors:** Michele Bevere, Anastasios Gkountakos, Filippo Maria Martelli, Aldo Scarpa, Claudio Luchini, Michele Simbolo

**Affiliations:** 1Department of Diagnostics and Public Health, Section of Anatomical Pathology, University and Hospital Trust of Verona, 37134 Verona, Italy; 2ARC-Net Applied Research on Cancer Centre, University and Hospital Trust of Verona, 37134 Verona, Italy

**Keywords:** pancreatic neuroendocrine tumors, functioning pancreatic neuroendocrine tumors, diagnosis, molecular alterations, treatment

## Abstract

Pancreatic neuroendocrine neoplasms (PanNENs) are rare neoplasms arising from islets of the Langerhans in the pancreas. They can be divided into two groups, based on peptide hormone secretion, functioning and nonfunctioning PanNENs. The first group is characterized by different secreted peptides causing specific syndromes and is further classified into subgroups: insulinoma, gastrinoma, glucagonoma, somatostatinoma, VIPoma and tumors producing serotonin and adrenocorticotrophic hormone. Conversely, the second group does not release peptides and is usually associated with a worse prognosis. Today, although the efforts to improve the therapeutic approaches, surgery remains the only curative treatment for patients with PanNENs. The development of high-throughput techniques has increased the molecular knowledge of PanNENs, thereby allowing us to understand better the molecular biology and potential therapeutic vulnerabilities of PanNENs. Although enormous advancements in therapeutic and molecular aspects of PanNENs have been achieved, there is poor knowledge about each subgroup of functioning PanNENs.Therefore, we believe that combining high-throughput platforms with new diagnostic tools will allow for the efficient characterization of the main differences among the subgroups of functioning PanNENs. In this narrative review, we summarize the current landscape regarding diagnosis, molecular profiling and treatment, and we discuss the future perspectives of functioning PanNENs.

## 1. Introduction

Neuroendocrine neoplasms (NENs) are a heterogenous group of rare tumors with different morphological features, immunophenotype, molecular profiling and clinical presentation that can virtually originate in every part of the body, including lung, small intestine and pancreas [1,2].

Pancreatic NENs (PanNENs) arising from the islets of Langerhans, account for 12% of all NENs and 1–3% of all types of pancreatic cancers [3,4].

Around 15% of cases secrete hormones leading to clinical symptoms and thus are known as functioning PanNENs. Based on the secreted peptide, the functioning PanNENs can be divided into insulinoma (insulin), gastrinoma (gastrin), glucagonoma (glucagon), somatostatinoma (somatotastin) and VIPoma (vasoactive intestinal peptide; VIP), along with other less common tumors producing serotonin and adrenocorticotrophic hormone (ACTH) [5,6,7]. However, most PanNENs (85%) are nonfunctioning because they do not secrete clinically significant levels of hormones. The nonfunctioning PanNENs are characterized by a worse prognosis, compared with hormone-secreting functioning PanNENs, due to a lack of symptoms which inevitably leads to a late diagnosis [5,6,7].

The current WHO’s classification [8] divides PanNENs into: well-differentiated (also known as PanNETs, that are further subdivided in grades 1–3), poorly differentiated (also known as PanNEC and further divided into small and large cells) and mixed neuroendocrine/non-neuroendocrine form (also known as MiNEN), as summarized in Figure 1. 

PanNETs are well-differentiated tumors and are classified in G1 (<2 mitotic count/mm^2^; <3% Ki67 index), G2 (2–20 mitotic count/mm^2^; 3–20% Ki67 index) and G3 (>20 both mitotic count/mm^2^ and % Ki67 index). While PanNECs are poorly differentiated tumors that are always characterized by the mitotic count and Ki67 index of >20 [8]. 

Clinically, survival worsens as the grade of differentiation increases [9,10,11].

## 2. The Molecular Landscape of Familiar and Sporadic PanNENs

The familiar syndromes account for 10% of all PanNENs and are characterized by an inherited detrimental germline mutation in a tumor suppressor gene, thereby causing an increased tumor susceptibility and tumor formation in the pancreas. The familiar syndromes include the multiple endocrine neoplasia type 1 (MEN1), von Hippel–Lindau disease (VHL), neurofibromatosis type 1 (NF1) and tuberous sclerosis complex (TSC) [1,2,12]. All of these germinal syndromes promote the activation of key pathways favoring tumor proliferation and growth.

*MEN1* encodes for a protein, menin, ubiquitously expressed mainly in the nucleus. Menin plays crucial roles in the regulation of the expression of cyclin-dependent kinase inhibitors through the interaction with *KMT2A/2D* (also known as *MLL1/2*), inhibition of PI3K/mTOR, activation of DNA repair genes (*BRCA1* and *RAD51*) and preservation of telomere stability [1,2,13]. However, about 30% of sporadic PanNENs harbor somatic *MEN1* mutations [14]. The *VHL* codifies for pVHL, a negative regulator of hypoxia-inducible factor (HIF) and the loss of pVHL induces an increase of vascular endothelial growth factor receptor (VEGFR) and platelet-derived growth factor receptor (PDGFR) [1,2]. The *NF1* gene product, neurofibromin, is a negative regulator of rat sarcoma (Ras)/mitogen-activated protein kinases (MAPK) and PI3K/mTOR signals. Consequently, the loss of neurofibromin promotes the activation of both RAS/MAPK and PI3K/mTOR [1,2]. *TSC1* and *TSC2* encode for hamartin and tuberin, respectively (negative regulators of PI3K/mTOR) [1,2].

Interestingly, Scarpa et al. found that in sporadic PanNETs, 17% of patients harbored germinal mutations in *CDKN1B*, *MUTYH*, *CHEK2* and *BRCA2* [15], while other studies revealed that few PanNETs were affected by germinal mutations involving *APC, RAD50, RECQL4, FANCC, MAPKBP1* and *PIF1* genes [13,16]. This suggests the presence of other PanNENs-predisposing germline mutations not directly related to clinical syndromes.

Concerning the sporadic PanNETs, most of our information concerning signaling network derived from two large-scale reports: a whole-exome sequencing study [17] and a whole-genome landscape/RNA-sequencing study [15]. First, Jiao et al. identified the key mutated genes (*MEN1, DAXX/ATRX, PTEN, TSC2*) in 68 resected and sporadic PanNETs (mostly G1/G2), thereby highlighting the main genetic differences between PanNETs and PDAC [17]. Second, Scarpa et al. confirmed this finding and revealed novel molecular events in 102 resected and sporadic PanNETs (mostly G1/G2), allowing to identify at least four functional altered pathways in PanNETs: DNA damage repair (*MUTYH, CHEK2, BRCA2*), chromatin remodeling *(MEN1, SETD2, ARID1A, MLL3, SMARCA4*), telomere alteration (*TERT*, *DAXX/ATRX*) and PI3K/mTOR signaling pathway (*EWSR* fusions, *PTEN, TSC2, TSC1, DEPDC5*) [15]. Although very rare events, chromosome rearrangements were also observed and can cause the inactivation of suppressor genes, including *ARID2*, *CDKN2A*, *SETD2*, and gene fusions, as *EWSR1-BEND2/FLI, TSC1-TMEM71* and *CHD7-BEND2* in PanNETs [15,18].

Metastatic forms were also analyzed in order to identify molecular mechanisms underlying molecular progression. A whole-genome sequencing performed on metastasis derived from 20 advanced or metastatic PanNETs (G2/G3), and showed mainly somatic aberrations in *MEN1, DAXX, DMD, SETD2, ATRX* and *CREBBP* [19]. From a transcriptomic point of view, single-cell RNA sequencing from primary and metastatic tissues of one patient with PanNET G2, revealed that metabolic reprogramming, hypoxia and cell proliferation pathways guide the tumor progression [20]; however, this study showed high intra- and inter- heterogeneities in the tumor microenvironment, such as the cancer-associated fibroblasts (CAFs) of PanNET harboring different gene expression profiles from those expressed in PDAC and myCAF/iCAF [21].

Another molecular mechanism of metastatization was identified in a recent study where it was observed that the immunohistochemical expression of P53 and RB1 was normal in NETs G1/G2, but abnormal in NETs G3. Noteworthy, these alterations in the protein expression have been found only in NET G3 metastasis and not in primary tumors [22,23,24]. This supports the hypothesis that the progression of PanNET G1/G2 to PanNET G3 is based on acquired genetic alterations driving the metastasizing process (i.e., *TP53* and *RB1*), particularly to the liver, as also observed in other types of NETs (gastro-entero-pancreatic and lung NETs) [22,23,24].

Few studies interrogated the genomic landscape of PanNETs G3. For a total of 44 PanNETs G3, three studies detected a few mutations affecting *MEN1, ATRX/DAXX* and, less frequently, *TP53*, *CDKN2A, ARID1A, LRP1B* and *APC* genes, without any *KRAS* mutation or Rb loss [25,26,27]. Another study showed only mutations in *ATM, VHL* and *IDH1* in 15 gastro-entero-pancreatic NET G3 [28].

Unlike PanNETs G3, PanNEC often showed Rb loss and mutations in *KRAS*, *TP53, BRAF, RB1, APC, MYC, ARID1A, ATM, KDM5A, ESR1, CDKN2A, ARID1A* and *LRP1B* [25,26,27,29,30]. Interestingly, only some differences found between small cell and large cell GEP-NEC, especially *BRAF*, *MYC* and *ARID1A* mutations, were more frequent in large-cell PanNEC [27], while Bcl-2 was overexpressed in small PanNEC [30]. Overall, PanNECs share only singular mutations in five different genes (*TP53, CDKN2A, ARID1A, LRP1B* and *APC*) with PanNETs G3, and thus is more similar genetically and phenotypically with PDAC [26]. These different genomic alterations between PanNETs G3 and PanNEC may explain why PanNETs G3 have a low response rate to platinum-based chemotherapy, compared to PanNECs (which are similar to PDAC) [11]. Accordingly, the loss of Rb immunolabeling and *KRAS* mutation demonstrated that they are valid predictors of the response to platinum-based chemotherapy and indeed specifically detected in PanNEC [25]. The current guidelines suggest using *MEN1*/*ATRX*/*DAXX* and *RB1*/*TP53* to discriminate between PanNET G3 and PanNEC [31,32]. However, Venizelos et al. highlighted that the Rb loss represents the better approach than *RB1* mutations to discriminate PanNET3 G3 versus PanNEC [27].

In 2015, Sadanandam et al. identified three transcriptomic subgroups in human PanNETs: islet/insulinoma, metastasis-like primary (MLP) and intermediate subtype [33]. The islet/insulinoma tumors are the less aggressive subtype characterized by the expression of insulinoma-associated genes (*INS, IAPP, INSM1*); the MLP subtypes are associated with a high rate of liver or lymph node metastasis and enriched for genes correlated with stroma, hypoxia and pancreatic progenitor-specific genes; the intermediate subtype is more similar to islet/insulinoma tumors but more aggressive with a high frequency of *MEN1* and *DAXX/ATRX* mutations [33]. These three subtypes were further confirmed by Scarpa et al. in 2017 [15]. Recently, a multi-omics approach of RNA-sequencing, global proteome profiling and whole-exome sequencing was used to characterize a non-selected group of PanNENs (*n* = 84). This study has found four different subgroups: proliferative, stromal/mesenchymal, alpha cell-like and PDX1-high [34]. The proliferative subgroup contains both well and poorly differentiated tumors and showed molecular features of cell cycle progression (enrichment of MYC targets, G2M checkpoint, E2F targets) [34]. Accordingly, this subgroup is associated with a worse prognosis, suggesting that the proliferative subgroup is more like PanNEC than other PanNETs, in terms of transcriptomic features. The stromal/mesenchymal subgroup showed both increased mRNA levels and relative activation of *YAP1* and *WWTR1* (the Hippo signaling pathway), thereby increasing the activation of epithelial-mesenchymal transition and angiogenesis [34]. The stromal/mesenchymal subgroup may be sensitive to the inhibition of *YAP1* and *WWTR1*. Alpha cell-like subgroup showed increased expression of transcription factor *ARX* and mitochondrial proteins (i.e., glutaminase and arginase 2), and is enriched by oxidative phosphorylation-related genes associated with frequent mutations in *MEN1*, *DAXX* or *ATRX* [34]. This may be sustaining the susceptibility of this subgroup to glutaminase inhibition. The two large-scale reports [15,17] have demonstrated that the major mutations in PanNETs affect tumor suppressor genes (i.e., *MEN1, ATRX, DAXX*) with a low incidence in proto-oncogenes. Strikingly, the PDX1-high subgroup showed high expression levels of *PDX1* and is associated with high frequency in proto-oncogenes, including *CTNNB1* (p.D32N), *HRAS* (p.Q61R), *NRAS* (p.Q61R), *KRAS* (p.L19F and p.Q22K) and *RET* (p.V292M) [34].

The epigenetic regulation of DNA via methylation is strongly associated to tumor initiation and development in several human cancer types [35]. The global methylation status is inversely correlated with the grade, with significantly higher DNA methylation in G1, compared to G2/G3 PanNETs [36]. Gene mutations in *SETD2, MEN1* and *ATRX* can influence histone deacetylases (HDACs) and DNA methylation [37,38,39]. Indeed, high levels of all classes of HDACs, especially HDAC5, are correlated with high grade PanNENs and a poor outcome [40]. Previous studies showed that genes, including *RASSF1, HIC1, APC, CDKN2A, MGMT, MLH1, TIMP3, BRCA1* and *VHL*, involved in key pathways (i.e., Wnt/β-catenin pathway, cell cycle regulation, angiogenesis, DNA repair) are hypermethylated in PanNENs [1,41,42,43]. Accordingly, promising results are observed in vitro using inhibitors of DNA methyltransferase (azacytidine) and histone deacetylation (butyrate, valproic acid, trichostatin A and MS-275), supporting the need to validate these positive effects further [44,45]. In addition, a recent study has interrogated the methylome in patients with sporadic disease or PanNETs with hereditary syndromes (MEN1 and VHL) using a genome-wide DNA methylation approach [46]. This study showed DNA hypermethylation in MEN1-related PanNETs and DNA hypomethylation in VHL-related PanNETs. Moreover, this study discovered *APC* promoter hypermethylation in MEN1-related PanNETs [46]. Based on methylome profiling of PanNETs, three subgroups are identified: α-, intermediate and β-like [47]. The α-like PanNETs are associated with the higher frequency of *MEN1* mutations, while the β-like one is associated with *MEN1/DAXX/ATRX* wild-type. Instead, the intermediate subgroup presents mutations in *MEN1* and/or *DAXX/ATRX* with an increase in copy-number variation (CNV) events, compared to the α- and β-like subgroups [47]. Moreover, this study unrevealed the cell tumor origin correlated with transcription factor expression using DNA-methylation profiling [47]. Indeed, α-like tumors expressed ARX, β-like tumors expressed PDX1 and intermediate PanNETs were positive mostly for ARX and in few cases negative for both PDX1 and ARX [47]. As the previous study [47], another report identified three subgroups, based on methylome profiling, named T1-3 [48]. The T1 subgroup encompasses functioning PanNETs with *ATRX, DAXX* and *MEN1* wild-type genotypes. The T2 subgroup is associated with tumors harboring mutations in *ATRX, DAXX* and *MEN1* and recurrent patterns of chromosomal losses. T3 subgroup includes PanNETs G1 tumors carrying mutations in *MEN1* and recurrent loss of chromosome 11; this subgroup is associated with a better prognosis [48]. Using the DNA methylation-based approach, another study has recently allowed for identifying the different cells of origin for both PanNECs (acinar-like tumors via SOX9 expression similar to PDAC) and for all grades of PanNETs (endocrine cell of origin signatures similar to α cells) [49], confirming further the assumption that PanNETs and PanNECs are two different entities.

Interestingly, different candidate protein biomarkers, such as FASLG, which is negatively correlated with Ki67 and found in lower levels in PanNETs G3 [50], could be useful for diagnosis, prognosis and the detection of therapeutic targets [51].

Finally, the analysis of spliceosomes in 20 PanNETs discovered the overexpression of *NOVA1,* which is highly associated with cell proliferation, invasion, and migration. This study shed the light on the role of splicing machinery in carcinogenesis, thereby paving the way for the development of a new class of biomarkers in PanNETs [52].

The information about the molecular landscape of sporadic PanNENs are summarized in Table 1.

## 3. Focus on Functioning PanNENs

### 3.1. Insulinoma

It is the most common functioning PanNEN and accounts for about 4–20% of PanNENs with four cases per 1 million person-years [53,54]. Insulinoma is characterized by the uncontrolled secretion of insulin from β-cells, thereby causing a hypoglycemic syndrome and, in turn, adrenergic symptoms (palpitations and tremor), cholinergic symptoms (sweating, hunger, and/or paraesthesia) and neuro glycogenic symptoms (a wide variety of psychiatric and neurological manifestations) [55]. About 10% of individuals with *MEN1* mutation develop insulinoma [56]. The best diagnosis is based on the hypoglycemic symptoms, low plasma glucose levels and symptom relief after glucose administration (the Whipple triad), documented by the finding of symptoms, signs, or both with plasma concentrations of glucose < 55 mg/dL (3.0 mmol/L), insulin ≥ 3.0 μU/mL (18 pmol/L), C-peptide ≥ 0.6 ng/mL (0.2 nmol/L), proinsulin ≥ 5.0 pmol/L, and the absence of sulfonylurea (metabolites) in the plasma and/or urine after 48–72-h fasting [57,58]. The tumors are usually detected by computed tomography (CT), magnetic resonance imaging (MRI) and endoscopic ultrasonography (EUS) and/or 68Ga-DOTATOC/TATE PET [57]. For all types of functioning PanNENs, a histological examination on HE-stained sections must be accompanied by immunostaining for neuroendocrine markers (synaptophysin, chromogranin A) and the specific hormonal syndrome suspected clinically. Both a mitotic index using a mitotic count and a Ki67 index are mandatory. The growth pattern of insulinomas is mainly trabecular or solid. Histologically, some insulinomas show a tubuloacinar growth pattern with psammoma bodies, as somatostatin producing PanNEN. For all PanNENs (both functioning and nonfunctioning), the prognostic markers are the surgical resection margin, G stage, TMN stage, lymph node, metastasis, vascular invasion and the necrosis [59]. Moreover, the loss of *DAXX* and/or *ATRX* is associated with an increased risk of metastasis [60]. In a multicenter study of 31 patients with malignant insulinoma (<10% of all diagnosed insulinomas), the 5-year and 10-year survival rate was 62% and 49%, respectively. The median overall survival was 40 months but increases significantly in patients with malignant insulinoma with a low grade tumor (G1) and low Ki67 (<10%) [61].

### 3.2. Gastrinoma

It accounts for 4–8% of all PanNENs [62] and is more common in the duodenum, typically localized in its proximal part [63]. Gastrinoma is characterized by uncontrolled gastrin secretion which leads to overproduction of gastric acid from G cells, thereby causing Zollinger–Ellison syndrome, the second most common hormonal syndrome associated with functioning PanNENs [64]. About 25–30% of gastrinomas form part of the inherited syndrome MEN1 [65]. Typical clinical features are duodenal ulcer and/or gastro-esophageal reflux disease, abdominal pain, and diarrhea [66]. The diagnosis is based on high levels of fasting gastrin, gastric pH (preferably ≤ 2) and common imaging tools [64]. Histologically, gastrinoma often displays trabecular or glandular structures, while the tumor margin can be expansive or focally infiltrative and the stroma is normally delicate. The cancer cells express synaptophysin (SYP), chromogranin A (CgA), somatostatin receptor 2 (SSTR2) and gastrin (often focal) [64]. Among 160 patients with gastrinoma, the 15-year disease-related survival was 98% for operated and 74% for unoperated, respectively [67]. Instead, in MEN1-related patients with gastrinoma, 5- and 10-year overall survival rates were 83% and 65%, respectively [68].

### 3.3. Glucagonoma

Glucagonoma predominantly involves the tail of the pancreas with an incidence rate of about two cases per 20 million person-years and accounts for 1–2% of all PanNENs [69,70]. It can be associated with uncontrolled glucagon secretion from cells producing glucagon and preproglucagon (α-cells), thereby causing glucagonoma syndrome [69]. Glucagonomas develop in less than 3% of subjects with MEN1 [56]. The diagnosis is based on high glucagon serum levels and the typical triad of glucagonoma syndrome: skin rash (necrolytic migratory erythema), diabetes mellitus and weight loss [71]. Histologically, glucagonoma shows densely packed trabecular formations and a scant stromal reaction. No poorly differentiated glucagonomas have been described, but glucagonomas may progress to G3. The tumor cells express SYP, CgA, glucagon and often pancreatic polypeptide (PP). In a retrospective study of 23 cases, no correlation between survival and the glucagon level or the Ki-67 index was found [72]. In an old study of 233 patients with glucagonoma, the 10-year survival rate was 51.6% with metastasis and 64.3% in those without metastasis, respectively [73]. In a more recent study of only six patients, 5-year survival was 66% [74].

### 3.4. Somatostatinoma

It represents 4% of PanNENs with an estimated incidence of one case per 40 million person-years with a female predominance (2:1) [75]. Somatostatinoma is often associated in the context of familiar syndromes, including MEN1, VHL and NF1; moreover, it is a frequent tumor type in the duodenum and rare in the pancreas [75,76]. In less of 10% of patients, somatostatinoma can be associated with clinical manifestations of inappropriate somatostatin secretion, thereby causing the classic triad of somatostatinoma syndrome (diabetes/glucose intolerance, cholelithiasis and diarrhea/steatorrhea) [75]. These tumors are mainly silent and diagnosed incidentally or present with non-specific symptoms, thereby leading to a challenging diagnosis [77]. The diagnosis is based on high fasting plasma somatostatin hormone concentration and common imaging tools [66], also including somatostatin receptor scintigraphy (OctreoScan) and functional positron emission tomography (PET) tracers, as 18F-DOPA and 11C-5-HTP [78]. A subset of pancreatic somatostatinoma exhibits a tubular and glandular architectural pattern and intraglandular psammomatous calcifications. However, the majority of somatostatinomas show the typical appearance of other PanNENs. The tumor displays diffuse positivity for SYP, somatostatin, and the less consistently positive or absent CgA. It may also show scattered positivity for PP, calcitonin, gastrin, ACTH, glucagon, and insulin. Large tumor size (>3 cm) and lymph node involvement are poor prognostic markers [79]. In an old study with 173 patients with somatostatinoma, the overall postoperative 5-year survival rate was 75.2%, but 59.9% and 100% in patients with and without metastasis, respectively [79]. In a more recent study of 11 patients, the overall survival was 47.7 months [75].

### 3.5. VIPoma

It accounts for 0.6–1.5% of all PanNENs with an incidence of one case for 10 million person-year, yet it is observed in approximately 5% of MEN1 patients [80]. VIPoma is usually associated with uncontrolled secretion of VIP by δ-cells, thereby causing watery diarrhea, hypokalemia, hypochlorhydria/achlorhydria and acidosis [81,82,83]. The diagnosis is confirmed by high plasma VIP levels associated with the common imaging tools, including PET and MRI [84,85]. Histologically, VIPoma resembles the features of other well differentiated functioning PanNENs with a lymph vascular and perineural invasion. Tumor cells express SYP, CgA, cytokeratin AE1/AE3, 8/18, 19. In a recent retrospective study, four patients were affected by VIPoma among 326 patients with PanNENs. Compared to the other functioning PanNENs, VIPomas were all located at the pancreatic tail, were larger with a higher Ki-67 index and more metastasis [86]. One study showed that the average survival rate of 18 patients with VIPoma was 96 months [87].

### 3.6. Serotonin-Producing Neuroendocrine Tumor

It accounts for 0.58–1.4% of all PanNENs and derives by uncontrolled proliferation of Kulchitsky cells or Enterochromaffin cells that express serotonin [88,89]. Most patients are diagnosed with liver metastasis [90]. The typical carcinoid syndrome includes abdominal pain, diarrhea, weight loss and flushing [91]. The diagnosis is confirmed by high urinary excretion of 5-hydroxy-indolacetic acid, the principal metabolite of serotonin. The cancer cells are frequently organized in a trabecular pattern and less frequently in solid nests with prominent stroma [92]. Unlike they rarely express substance P, acidic fibroblast growth factor and CDX2, they usually present the immunohistochemical expression of serotonin and somatostatin receptor 2a [88]. In an old study of 46 patients with serotonin-producing tumor, the 5-year survival rate was low (around 30%) [93].

### 3.7. ACTH-Producing Neuroendocrine Tumor

It is an extremely rare tumor characterized by uncontrolled production of ACTH from bronchial carcinoid cells. This, in turn, can lead to the increase in glucocorticoid levels and, consequently, to the Cushing syndrome [94,95]. Occasionally, Cushing syndrome and Zollinger–Ellison syndrome can show up together [96]. The Cushing syndrome presents a wide spectrum of symptoms, such as weight gain, central obesity, moon face, hypertension, insulin resistance and glucose hypersensitivity [97]. The diagnosis is based on 24-h urinary cortisol determinations and serum cortisol assessment after dexamethasone suppression combined with common imaging studies. Histologically, no morphological feature may distinguish ACTH-producing tumor from the other functioning PanNENs. The tumor cells are positive for ACTH, CgA, CD56, but negative for insulin and gastrin [94,98]. About 80% of patients present with metastasis (especially in the liver) at the time of diagnosis or progressively develop distant metastasis during follow up [94]. In a study of 11 cases, at 5- and 10-years after diagnosis, 35% and 16.2% of patients were alive, respectively [99].

## 4. Molecular Alterations in Functioning PanNENs

Given their low incidence, all the available information regarding genetic alterations of functioning PanNENs are derived from low-scale reports or case reports.

According to recent studies on sporadic insulinoma, *YY1* (T372R) mutation is predominant (30%) [100,101,102] but somatic *MEN1* mutations occur rarely (7%) [103]. As mentioned in the familial PanNENs, *MEN1* is a tumor suppressor, and it interacts with several proteins, receptors, and transcriptional factors [1,2,13]. Among more important, the loss of menin induces activation of Wnt/β-catenin signaling and TGF-β via SMAD3 interaction and suppresses the transcriptional activity of JUND (thus promoting cell proliferation) of specific genes regulating cell cycle, methylation, DNA repair and telomere stability via *DAXX*, histone methyltransferases (MLL1-2), *BRCA1/RAD5 1* [104]. The YY1 protein can increase the mTOR pathway and insulin secretion from β cells and their proliferation [105]. The mTOR/P70S6K activation is observed is higher compared to normal tissue and thus the mTOR inhibitors restored the proliferation in vitro [106]. The mTOR signaling pathway and its downstream serine/threonine kinase p70S6k lead to promote cell growth and G1 cell cycle progression [107]. Moreover, two different insulinoma subtypes have been described and named as CNV neutral and CNV amplification. The first subgroup showed a high rate of *YY1* mutations and loss of chromosome 11, while the second one had nearly no *YY1* mutations and gains of chromosomes 7, 3p, 5p and 13q [108,109]. In another study, the loss of heterozygosity (LOH) of chromosome 1q (1q 21.3-23.2 and 1q31.3), was frequently observed in sporadic insulinomas and LOH 1q21.3-23.2 was associated with insulinoma development [110]. Several tumor suppressor genes are in this part of chromosome 1, including HRPT2, MDA7/IL-24, IFI16, and thus promoting tumor proliferation and growth [110]. Moreover, chromosomal alterations and loss of telomeric ends were strongly associated with metastatic disease [111]. It has also been reported that *MAFA* missense mutation drives the development of familial insulinomatosis, characterized by the synchronous and metachronous occurrence of insulinomas, multiple MEN1-associated insulinoma precursor lesions and the rare development of metastasis [112]. *MAFA* regulates insulin expression and other genes involved in glucose-stimulated insulin secretion. This missense mutation promotes β cells proliferation [112]. Based on transcriptomic profiling, three molecular and clinically different subgroups of duodeno-pancreatic neuroendocrine tumors, including 31 functioning PanNETs (mostly insulinomas), were identified [113]: (1) better-outcome subgroup with a mature beta-cell phenotype, mostly insulinoma; (2) intermediate-outcome subgroup with pancreatic progenitor and exocrine differentiation (mostly PanNETs G1 and G2); (3) and poor-outcome subgroup with greater dedifferentiation with hepatic and pancreatic alpha signatures, including different types of NENs (mostly PanNETs G3 and PanNECs) [113]. Insulinoma showed a major rate of hypermethylation among the other subgroups of functioning PanNENs, especially in epigenetic modifying enzymes as *INS/IGF2* locus, *CDNK1C*, *MEN1, KDM6A, MLL3/KMT2C, YY1, KDM5B,* and *SMARCC1* [114,115].

In gastrinoma, a high rate (about 40%) of somatic *MEN1* mutations [103,116,117] and deletions in chromosome 1q [118] are reported. Although large deletions or amplifications are relatively rare, gastrinoma is characterized by amplification of the *HER-2*/*neu* proto-oncogene or chromosome 9q, deletion of the *p16/MTS1* tumor suppressor gene or deletion of chromosome 3p [119]. Amplification of HER2 receptor triggers signaling cascades leading to activation of key pathways involved in tumor development, such as Ras/MEK/ERK, JAK/STAT, and PI3K/AKT [120]. The protein p16 is encoded by the cyclin-dependent kinase inhibitor 2A (CDKN2A) or multiple tumor suppressor 1 (MTS1) gene. The p16 protein binds to CDK4 or CDK6 and inhibits the formation of the complex between CDK4 or 6 and cyclin D. The absence of this complex formation retains the retinoblastoma protein and, in turn, leads to G1 cell cycle arrest. The p16 protein acts as a tumor suppressor and thus its loss via by homozygous deletion or promoter methylation and point mutation represents a key event to tumor development [121]. Moreover, gastrinoma exhibited a high number of hypomethylated genes codifying for metalloproteinases and members of the serpin family [114]. However, the methylation of tumor suppressor *CDKN2A* gene is the most common epigenetic alteration observed in gastrinoma [122].

Regarding the molecular profile of glucagonoma, few studies have been performed. In these studies, it was observed that glucagonoma is characterized by a high rate of *MEN1* mutations, including *MEN1* E179V [123] and two novel *MEN1* mutations, one heterozygous mutation 928G>C (G310R) in exon 7 [124] and the second a missense mutation at codon 561 in exon 10 (M561R9) [125]. Recently, a whole-exome sequencing of a patient affected by glucagonoma with necrolytic migratory erythema detected a biallelic inactivation of *DAXX* [126]. It forms a complex with ATRX and is necessary for H3.3 deposition at telomeres and pericentric heterochromatin promoting telomere stability [127]. Moreover, a case report described only glucagon receptor gene mutations without genetic alterations in *MEN1* or *VHL* [128]. This mutation leads to altered glucagon signaling promoting glucoganoma development [128]. To date, no further molecular characterization has been performed. 

In addition to the high rate of *MEN1* mutations [129], *HIF2A* somatic mutations were found in somatostinoma, as well as in other rare neuroendocrine tumors affecting other organs, such as paragangliomas [130]. In particular, the gain-of-function *HIF2A* is associated with a new syndrome of paraganglioma and somatostatinoma associated with polycythemia [131,132]. Under normoxic conditions, VHL binds HIF2A and, thus, promotes its degradation via ubiquitination. Under hypoxia, this regulatory mechanism is suppressed, so HIF2A is not degraded and can activate the transcription of genes involved in survival and cell proliferation (through the inhibition of p53 and activation of c-Myc), vascularization and metastasis (via VEGF activation) [133]. For this functioning subtype, no further genetic investigations were available.

Our knowledge about the molecular landscape of VIPoma is related to the low number of case reports, most of them reported *MEN1* mutations [103,129]. In addition, one case report observed a defect in the mismatch repair system gene *MSH2* and overexpression of CXCR4 in the hepatic metastasis [134]. Mutations in *MSH2* promote defects in the mismatch repair system and, thus, tumor development [134].

Concerning serotonin-producing tumors, a recent study revealed that they are characterized by few pathogenic mutations and that TGF-β pathway activation signatures were associated with extracellular matrix remodeling [135]. TGF-β receptor is activated by its ligands and, thus, triggers activation of the intracellular effectors, SMADs, thereby inducing transcription of their target genes involved in fibrosis, metastasis and tumor proliferation [136]. This may explain the reason why serotonin-producing neuroendocrine tumor is characterized by high desmoplastic stroma and a high rate of liver metastasis.

Finally, in a series of seven ACTH-producing tumors, the hypomethylation in pro-opiomelanocortin promoter was found leading to the ACTH abnormal releasing [137]. No further genetic investigations were performed.

The aforementioned mutations found in each subgroup of functioning PanNENs are summarized in the following table (Table 2).

## 5. The Therapeutical Options for Functioning PanNENs 

The surgical resection remains the only possible treatment for patients with localized functioning PanNENs without widespread metastasis [139,140,141]. If patients with functioning PanNENs are diagnosed in an advanced state with metastasis, the surgery option is not considered [142]. 

However, all patients with metastatic functioning PanNENs and without contraindication should have surgical exploration with the aim to remove the primary tumor and peritumoral lymph nodes [143].

The first consideration in the management of patients with advanced and unresectable functioning PanNENs is hormonal control to improve survival and quality of life [144]. The somatostatin analogues lanreotide, octreotide, pasireotide and their long-acting release forms are used to reduce the secretion of several hormones (especially in patients with VIPomas and glucagonoma) [145]. Together with somatostatin analogues, there are other specific cures used for one or more tumor types of functioning PanNENs. Indeed, diet intervention and diazoxide are used for the management of hypoglycemia in patients with insulinoma [145]. For patients with gastrinoma, high doses of proton pump inhibitors and H2 receptor blockers are recommended to reduce the high levels of acid secretion [145]. For patients with glucagonoma, somatostatinoma and VIPoma, diet intervention, including vitamin supplementation (especially for glucagonoma and somatostatinoma), and glucose control (especially for VIPoma), are recommended [145]. Despite that it may be present in other tumor types of functioning PanNENs, persistent diarrhea is the main clinical symptom. Telotristat ethyl, a tryptophan hydroxylase inhibitor, was approved to reduce bowel movements and diarrhea [145,146].

Regarding therapies that directly control tumor growth, the systemic treatment is indicated when unresectable and multiple metastasis are present. The somatostatin analogues are considered the first choice for low grade advanced functioning PanNENs (<10% Ki67) with a positive expression of somatostatin receptor [141,147,148,149]. This approach is the standard therapy in well-differentiated, locally advanced or metastatic functioning PanNENs of any size, except for insulinomas, gastrinomas and ACTH-producing tumors [150,151].

When patients progress under somatostatin analogues treatment or as a first-line treatment in patients with tumor-negative expression of somatostatin receptor, chemotherapy [152] or targeted therapies are used. There are few pharmacological schemes, based on temozolomide, streptozotocin (streptozocin mono- or plus 5-fluroracil) and platinum (cisplatin plus etoposide) [153]. Although their use is limited, chemotherapy is the standard of care for more aggressive PanNETs and PanNECs, with more positive effects in the second type of tumors [145,154]. Instead, for the other less aggressive PanNENs, two targeted drugs have been approved in patients with advanced functioning PanNENs, sunitinib (tyrosine-kinase inhibitor) and everolimus (mTOR inhibitor), showing an important progression-free survival [141,155,156].

In addition, for advanced functioning PanNENs with high levels of somatostatin receptors, the peptide receptor radionuclide therapy might be a sustainable option. Indeed, NETTER-1 trial has provided encouraging evidence for the treatment in advanced PanNENs, but this approach should be further validated in other trials [157,158].

To date, two clinical trials are recruiting for patients with functioning PanNENs, a phase III trial investigating the effects of cabozantinib, a tyrosine-kinase inhibitor (NCT03375320), and one study evaluating the possibility and the safety of radiofrequency ablation of the tumor under ultrasonography guidance (NCT03834701).

Immunotherapy is positively changing the treatment of some solid tumors, including lung cancer and melanoma [159], suggesting a potential role also in other solid tumors. The immunotherapy for PanNENs is still in the early stage of investigation and this field of research is investigating without discriminating functioning PanNENs from nonfunctioning ones. Indeed, although the microsatellite instability seems to show a low activity, the expression of programmed death-ligand 1–2 (PD-L1-2) and anti-programmed cell death protein 1 (PD-1) is found significantly correlated with shorter patient survival, high grade (G3) and aggressive PanNENs [160,161]. This led to performing clinical trials with the aim to investigate the therapeutic potential of immune checkpoint inhibitors. To date, there are only few clinical trials demonstrating moderate clinical benefit [162]. However, the combination immunotherapy (Ipilimumab and Nivolumab, anti-CTLA-4 and anti-PD-1 agents, respectively) showed a promising clinical activity in high-grade PanNENs. Indeed, three of seven patients with advanced PanNENs showed a high response rate [163]. In line with this, a recent study discovered that the MLP-1 subtype of PanNETs presents an overexpression of PD-L1, PD-L2, suggesting that immunotherapy may have clinical benefits, especially in patients with advanced PanNETs [164].

Overall, immunotherapy remains a promising therapeutic approach for the treatment of advanced PanNENs [165,166,167].

In addition, novel immunotherapeutic approaches using chimeric antigen receptor (CART)-cells demonstrated promising positive results in vitro and in vivo in PanNETs [168].

## 6. Discussion and Future Perspectives

The functioning PanNENs are a heterogeneous group of tumors with different morphogenic and clinical features [1,2]. Due to their low incidence, many studies are being performed on a small number of cases or pooled from different types of functioning PanNENs. In addition, for some rare subgroups of PanNENs, such as VIPoma and somatostatinoma, large-scale data are completely lacking. Noteworthy, in patients under 20 years of age, the functioning PanNENs represent 30% of all NENs, especially insulinoma and gastrinoma [169,170]. This emphasizes the necessity of multi-institutional collaborations to increase the knowledge about PanNENs.

The basic and preclinical research allowed us to better characterize the main pathways underlying the functioning PanNENs. Among all mice models [171], it is worth mentioning the RIP-Tag and menin-deficient mice, which contributed to test the efficacy of pasireotide, sunitinib and mTOR inhibitors [172,173,174], and a new mouse model harboring thymidylate synthase (TS) overexpression and *Men1* inactivation in pancreatic islet cells (*hTS/Men1*^−*/*−^). This mouse model allowed us to identify a crucial dualism between these two proteins in exacerbating PanNETs progression [175].

Table 3 summarizes the most common and new mice models used to decipher PanNENs.

To date, a study incorporating human tumor tissues, in vivo models and large organoid biobank [180,181] could be helpful to better understand the functioning PanNENs.

Moreover, the development of high-throughput techniques has clearly accelerated the research on PanNENs, allowing us to identify the common genetic and epigenetic alterations and paving the way for the introduction of targeted therapy, even if for a small number of patients [1,2].

Recently, a novel multi-gene liquid biopsy, based on real-time PCR, the NETest, has been demonstrated to be a reliable and accurate tool for the diagnosis, grading, staging, progression, and therapy responses of PanNENs [182,183,184,185,186]. The mRNA is isolated from EDTA-collected whole blood samples and real-time PCR is performed to interrogate 51 genes assessed by four different prediction algorithms. Then, a score (0–100) is generated to define the tumor activity and to provide more direct information about the tumor, its pathophysiology, and its tumor grade or progression [187]. The technique, based on gene expression-PCR provides a higher sensitivity to identify the molecular detection of microscopic diseases and early metastatic disease, compared to imaging or two clinically approved assays for CgA measurement in the blood (NEOLISA and CgA ELISA) [188]. In the detection of biochemical recurrence, the NETest was more accurate (84%), compared to the used diagnostic marker, CgA (51–57%) [188]. The decrease in NETest levels after surgical resection provided a precise tool for early assessment of surgical efficiency, whereas CgA did not show a clinical utility [189]. Noteworthy, the NETest was able to discriminate better G1 and G2 PanNENs, in comparison to CgA assays [188]. Overall, given the controversial usefulness of diagnostic tool CgA [190], the NETest should be included in clinical practice. 

In order to improve the clinical management of functioning PanNENs, a set of important initiatives should be taken in a five-year perspective plan. Liquid biopsy may be considered as a promising surrogate for tissue biopsy, allowing for the longitudinal monitoring of the disease in a non-invasive manner. This approach could significantly help towards early diagnosis, assessment of treatment efficacy, as well as detection of early onset of resistance, thus allowing adjustment of the treatment [191]. Although preliminary findings suggest the potentially high utility that liquid biopsy could exert, its role in functioning PanNENs is not sufficiently explored, except for the NETest [192,193,194,195]. Therefore, we strongly suggest the in-depth study of other aspects of liquid biopsy, such as circulating tumor DNA (ctDNA) in functioning PanNENs and interpretation of the findings with the NETest approach towards generating novel companion diagnostic tests (Figure 2).

Today, targeted therapy in functioning PanNENs is limited only to very few agents (everolimus, sunitinib). However, it is worth stating that this targeted approach is not chosen based on the molecular profiling of the patients. Interestingly, genetic profiling of different functioning PanNENs has revealed different actionable alterations, such as *HER-2* amplification in gastrinoma that could be the subject of targeted therapy (i.e., trastuzumab, pertuzumab) [119]. Consequently, we highly recommend the genomic profiling of every case of functioning PanNENs, not only to characterize the mutational landscape of each functioning subtype, but also to explore the possibility of treating the patient with a targeted agent (Figure 2). Given the important results of immunotherapy in other solid tumors, it would be important to design clinical trials enrolling patients with functioning PanNENs treated either with immunotherapy alone or in combination with chemotherapy.

However, there are different limitations in the studies on functional PanNENs. They are rare (about 15% of PanNENs) and heterogenous types of tumors, which make it difficult to recapitulate their features in vitro and in vivo. Most preclinical models (Table 3) develop insulinoma, which is the most common subtype of functional PanNENs, and thus the most studied.

Indeed, the main studies deciphering the molecular, transcriptomic, and epigenetic profiles of PanNENs (Table 1) have been investigated on mainly nonfunctioning tumors and insulinomas. 

Furthermore, the previous successful clinical trials [148,149,155,156] have studied both nonfunctioning and functioning (mainly insulinomas) NENs of different primary location sites (i.e., pancreas, lung, pituitary gland). This happens because the recruitment of a randomized study from an already limited patient population, is usually challenging both for appropriate statistical analysis and independent validation. Consequently, there are few findings for the other less common subtypes of functioning PanNENs regarding both molecular and therapeutical aspects.

To solve this gap of information in the future, we highly encourage the establishment of international multi-institution consortiums in order to share knowledge and increase the available samples for the research. This will translate into the recruitment of more patients for conducting clinical trials for each subgroup of each primary tumor site to decipher selectively the molecular features (Figure 2).

## Figures and Tables

**Figure 1 biomedicines-11-00303-f001:**
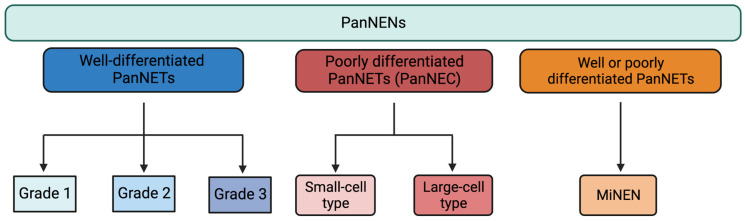
The last WHO’s classification of PanNENs (2019). PanNENs: pancreatic neuroendocrine neoplasms; PanNET: pancreatic neuroendocrine tumor; PanNEC: pancreatic neuroendocrine carcinoma; MiNEN: mixed neuroendocrine/non-neuroendocrine neoplasm.

**Figure 2 biomedicines-11-00303-f002:**
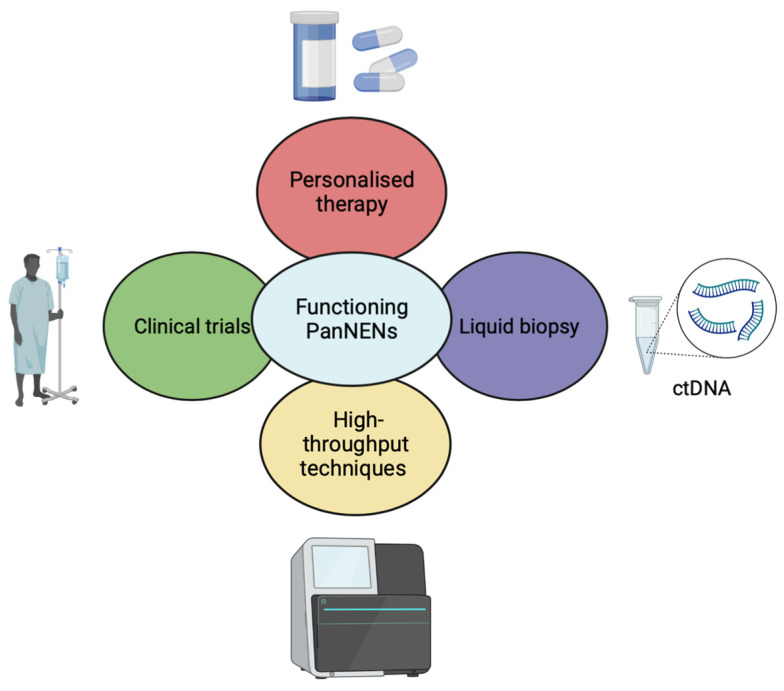
Future perspectives in functioning PanNENs. The multidisciplinary approaches, based on high-throughput techniques and recent tools (i.e., NETest, liquid biopsy) might lead to the identification of targeted molecular alterations in functioning PanNENs. PanNENs: pancreatic neuroendocrine neoplasms; ctDNA: circulant tumor DNA.

**Table 1 biomedicines-11-00303-t001:** The genomic, transcriptomic, and epigenetic landscape in sporadic PanNENs. MLP: metastasis-like primary, CNV: copy-number variation.

Genomic Landscape			
Classification	Differentiation	Gene Mutations	References
PanNET G1/G2	well-differentiated	*MEN1, DAXX/ATRX, PTEN, TSC2, MUTYH, CHEK2, BRCA2, SETD2, ARID1A, MLL3, SMARCA4, TERT, EWSR* fusions, *PTEN, TSC1, DEPDC5*	[15,17]
PanNET G3	well-differentiated	*MEN1, ATRX/DAXX, TP53*, *CDKN2A, ARID1A, LRP1B*, *APC*	[25,26,27]
PanNEC	poorly differentiated	*KRAS*, *TP53, BRAF, RB1, APC, MYC, ARID1A, ATM, KDM5A, ESR1, CDKN2A, ARID1A, LRP1B*	[25,26,27,29,30]
**Transcriptomic Landscape**			
**Classification**	**Subgroups**	**Molecular Findings**	**References**
Sadanandam et al. (2015); Scarpa et al. (2017)	islet/insulinoma	expression of insulinoma-associated genes	[15,33]
	MLP	expression of stroma-, hypoxia- and pancreatic progenitor-specific genes	
	intermediate subtype	*MEN1*, *DAXX/ATRX* mutations	
Yang et al. (2021)	proliferative	enrichment of MYC targets, G2M checkpoint, E2F targets	[34]
	stromal/mesenchymal	Hippo signaling pathway activation	
	alpha cell-like	high expression of *ARX* and mitochondrial proteins, enriched by oxidative phosphorylation-related genes associated with frequent mutations in *MEN1*, *DAXX* or *ATRX*	
	PDX1-high	high expression levels of *PDX1* associated with mutations in *CTNNB1*, *HRAS*, *NRAS*, *KRAS*, *RET*	
**Epigenetic Landscape**			
**Classification**	**Subgroups**	**Molecular Findings**	**References**
Di Domenico et al. (2020)	α-like	*MEN1* mutations, high expression of ARX	[47]
	intermediate	*MEN1* and/or *DAXX/ATRX* mutations with increased CNV, positive mostly for ARX or negative for both ARX/PDX1	
	β-like	*MEN1/DAXX/ATRX* wild-type, high expression of PDX1	
Lakis et al. (2021)	T1	*MEN1/DAXX/ATRX* wild-type	[48]
	T2	*ATRX, DAXX*, *MEN1* mutations and recurrent chromosomal losses	
	T3	*MEN1* mutation and recurrent loss of chromosome 11	

**Table 2 biomedicines-11-00303-t002:** The reported molecular alterations of each subgroup of functioning PanNENs with clinical presentations and frequency of MEN1-associated syndrome. LOH: loss of heterozygosity; VIP: vasoactive intestinal peptide; ACTH: adrenocorticotrophic hormone.

Tumor Types	Syndrome Related	Clinical Presentations	Molecular Alterations	References
Insulinoma	10% [56]	hypoglycemic symptoms	*YY1*, *MEN1*, mTOR/P70S6K activation, LOH chromosome 1q, *MAFA*, epigenetic dysregulation (*INS/IGF2* locus, *CDNK1C, MEN1, KDM6A, MLL3/KMT2C, YY1, KDM5B*, and *SMARCC1*)	[100,101,102,103,106,110,112,114,115]
Gastrinoma	25–30% [65]	esophageal symptoms, abdominal pain and diarrhea	*MEN1*, deletions in chromosome 1q, amplification of the *HER-2*/*neu* or chromosome 9p, deletion of the *p16/MTS1* or chromosome 3p, hypomethylated genes (metalloproteinases and serpin), methylation of *CDKN2A*	[114,116,117,118,119,122]
Glucagonoma	<3% [56]	skin rash, diabetes mellitus and weight loss	*MEN1* E179V and two novel *MEN1* mutations (G310R and M561R9), biallelic inactivation of *DAXX,* glucagon receptor gene mutations	[123,124,125,126,128]
Somatostatinoma	45% [138]	diabetes/glucose intolerance, cholelithiasis and diarrhea/steatorrhea	*MEN1, HIF2A*	[129,130,131,132]
VIPoma	5% [80]	watery diarrhea, hypokalemia, hypochlorhydria/ achlorhydria and acidosis	*MEN1, MSH2*	[129,134]
Serotonin-producing tumors	Rare (N/A) [138]	abdominal pain, diarrhea, weight loss and flushing	Low mutation drivers, TGF-β pathway activation signatures associated with extracellular matrix remodeling	[135]
ACTH-producing tumors	Rare (N/A) [138]	weight gain, central obesity, insulin resistance and glucose hypersensitivity	Hypomethylation in pro-opiomelanocortin promoter	[137]

Note: The reported information includes only somatic molecular alterations found in each tumor type.

**Table 3 biomedicines-11-00303-t003:** The most common and new mice models to study PanNENs. TS: thymidylate synthase; Pc2: prohormone convertase-2.

Mouse Models	Mechanism	References
RIP-Tag	The RIP1-Tag2 line develops insulinomas and was generated by cloning a known oncogenic driver (SV40) downstream of the rat insulin promoter for expression in β-islet cell. This model was served to demonstrate new therapeutics, such as sunitinib and mTOR inhibitors.	[172,173]
Menin-deficient mice	The menin-deficient mice developed PanNENs and it was widely used to test the efficiency of several treatments, including pasireotide).	[174]
*hTS/Men1* ^−*/*−^	Thymidylate synthase (TS) plays a crucial role in the early stages of DNA biosynthesis and its inhibition causes DNA damage. Elevated TS showed a pro-tumorigenic role in PanNETs. To better investigate these findings, a mouse model was generated where TS overexpression cooperates with Men1 inactivation in pancreatic islet cells (hTS/Men1^−*/*−^). This new mouse model showed that TS overexpression cooperates with Men1 deletion and favors the progression of PanNET and is associated with reduced survival rate.	[175]
INS-p25OE	This is a dox-inducible and conditional mouse model in which activation of the Cdk5 pathway in β-islet cells ensures to obtain a heterogenous series of tumors, both functioning (mostly insulinoma) and non-functioning PanNENs.	[176]
*pIns-c-MycER^TAM^/RIP-* *Bcl-x_L_*	To explore the consequences of activation of c-Myc when apoptosis is suppressed, a double transgenic model crossing a mice model of switchable c-Myc expression in pancreatic β cells under the control of an insulin promoter (*pIns*) and a mice model expressing Bcl-x^L^, under the direction of the rat insulin promoter (RIP7) has been generated. Bcl-x^L^ suppresses the mitochondrial apoptotic pathway, thereby blocking the Myc-induced apoptotic pathway. This model developed rapidly angiogenic, invasive islet tumors.	[177]
*Pc2* ^−*/*−^	Prohormone convertase-2 (Pc2) is an enzyme that plays an important role in the first step of glucagon synthesis. The Pc2 knockout (*Pc2* ^−*/*−^) mice developed an inability to covert proglucagon into glucagon, reduced plasma glucose, hyperplasia and tumor affecting α-cells. In conclusion, the blockage of the glucagon signal results in tumorigenesis.	[178]
MPR *(Men1^flox/flox^ Pten^flox/flox^* RIP-Cre*) and* MPM *(Men1^flox/flox^ Pten^flox/flox^* MIP-Cre*)*	Using the Cre-LoxP system, two mice models with insulin-specific biallelic inactivation of *Men1* and *Pten* were generated. The Cre in the MPR mouse model was driven by the transgenic rat insulin 2 promoter, while in the MPM mouse model was driven by the knock-in mouse insulin 1 promoter. These models developed rapidly more aggressive G1/G2 PanNETs. Accordingly, mTOR inhibition with rapamycin delayed the growth of PanNETs in both models.	[179]

## Data Availability

Not applicable.

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
