# Peer review of "An Insight on Functioning Pancreatic Neuroendocrine Neoplasms"

_biomedicines, 2023, doi:10.3390/biomedicines11020303_

Round 1

Reviewer 1 Report

The review ‘An Insight on Functioning Pancreatic Neuroendocrine Tumors’ is attempting to provide insight about pancreatic neuroendocrine tumor and its subtypes. The purpose of this paper is to summarize the current landscape of diagnosis, treatment, and molecular profiling of NENs as well as discuss the future perspectives of functioning NENs. Even though this review is well written, there are a few modifications that could be made to enhance its readability.

I have a few criticisms and suggestions to improve the review.

In certain points, this review provides vague ideas that do not provide explanations. For instance, the line 390 - 391 states that: ‘Recently, a novel multi-gene liquid biopsy based on real-time PCR, the NETest, is shaking the field of all types of NENs. However, in the line that mentions NETest, there is no explanation as to how and why the test is a reliable method for diagnosing, staging, and predicting the outcome of treatment. Cite a few examples and make your concept clear. Furthermore, please refrain from using the term "shaking" as it sounds unscientific.

The review can be improved by providing more information in the form of figures or lists. For example, the authors can provide the functions of the genes mutated in panNEN which is mentioned in Table 2 as a separate column in the same table or as a different table.

The review lacks information regarding all mice models that contributed to understanding PanNEN. For example, there was no mentioning about the mice model pIns-c-MycERTAM/RIP-Bcl-xL (PMID: 12015982) or Pc2 −/− (PMID: 24456331) in this review. Please mention about such mice models and explain how they contributed in obtaining novel information about pancreatic neuroendocrine tumor. Similarly, novel information from mice models is also generated recently. For example, Thymidylate synthase is cooperating with Men1 deletion for the progression of PanNET (PMID: 36048542). At the same time, the loss of PTEN can also cooperates with Men1 in the progression of PanNET (PMID: 31160716). Please include such recent and established information about PanNET in this review. The mice models should be provided as a list and insights obtained from each mouse model should be provided as separate columns in the list.

Author Response

We thank all reviewers for the time and effort to review our manuscript. The constructive criticism is highly appreciated and will certainly contribute to improve the quality of this manuscript. We hope that our discussion clarifies some concerns of the reviewer and again, want to thank you for the insightful review.

Here, please find the answers to the more common concerns.

Reviewer 2 Report

Specific comments to the authors

The submitted review “An Insight on Functioning Pancreatic Neuroendocrine Tumors” gathers, summarize and analyses heterogeneous aspects of the role of functioning panratice neuroendocrine neoplasm/tumors (PanNENs/PanNETs) on the basis of already published reviews and clinical trials as well as experimental investigations. The authors summarized and discussed clinical, hereditary and molecular aspects on functioning PanNENs. The submitted manuscript gives an interesting survey of functioning PanNENs, which is easy to read, to follow and to understand. The authors should clarify some aspects before accepting the manuscript for publication as mentioned below.

# "2. The molecular landscape of known and sporadic PanNENs": the authors should clearly separate the molecular findings of well-differentiated and poorly differentiated PanNETs in a separate table to show the significant changes. In addition, the epigenetic findings of the molecularly defined subsets of PanNENs (α-like tumors expressing ARX, β-like tumors expressing PDX1, and intermediate PanNETs) should also be listed in an additional table.

# "3. Focus on functioning PanNENs": authors should summarize the key features (in terms of clinical presentation, heredity, molecular findings/changes, and prognosis and prognostic markers) of all functioning PanNENs in one table.

# "4. The therapeutic options for PanNENs": related to the sentence "This led to the conduct of clinical trials aimed at investigating the therapeutic potential of immune checkpoint inhibitors." The authors should specify these clinical trials in more detail.

# "5. Molecular changes in functioning PanNENs": It is not clear why this chapter does not directly follow the chapter "3. Focus on Functioning PanNENs."

# "6. Discussion and Future Perspectives": The chapter and especially Figure 2 are largely superficial. Therefore, the authors should provide an outlook for the next five years on what approach should be taken to improve diagnostic and especially therapeutic strategies in the case of functioning PanNENs.

Author Response

(The authors gave the same response as above.)

Reviewer 3 Report

In their manuscript “An Insight on Functioning Pancreatic Neuroendocrine Tumors”, Bevere and Co-Authors aimed to:

1) summarize the current landscape regarding diagnosis, treatment, and molecular profiling, and  

2) discuss the future perspectives of functioning pancreatic neuroendocrine neoplasms (PanNENs).

ABSTRACT

1) The Authors divide PanNENs into two groups, functioning and nonfunctioning PanNENs, based on peptide secretion.  However, since they include in this classification serotonin-producing PanNENs, this classification is incorrect, and thus needs to be redefined.

2) The Authors state that “to date, only some therapeutic and molecular aspects of PanNENs have been defined”. Data from scientific literature contradicts these statements, given the enormous advancement in the therapy of PanNENs.  As for the advancement of molecular aspects, even with the consciousness of being far away from a “complete” knowledge, enormous progresses have been made, and they are indeed reported in different sections of the manuscript, with pertinent references.

MAIN TEXT

1) Page 2 is extremely clear, including table and figure, but is nothing more than a duplication of the 2019 WHO classification of PanNENs. Given the aims of the manuscript, and the subsequent background of the potential Reader, this page could be probably omitted or reduced.

2) Page 2 (line 69) to page 5 (line 217): this part, which is indeed a true masterpiece in the field of NENs, is unfortunately not related with the topic: functioning PanNENs.  A mention to the term “islet/insulinoma” PanNENs is indeed present, but only in 3 out of the 148 lines (page 3: lines 147, 148, and 152).

3) Page 5, Insulinoma: the description of disease suspicion, diagnostic methods, and tumor site detection are, in my opinion, too generic and vague. The survival rate reported is not the survival rate of general patients with insulinoma, but the survival rate of patients with malignant insulinoma. Line 236 can be then misleading.

4) Gastrinoma, Somatostatinoma, VIPoma, ACTHoma:  some points raised for Insulinoma, for the clinical and diagnostic part.

5) Serotonin-producing PanNEN: The sentence “Most patients present the atypical syndrome” is not supported by any reference. The description of the Atypical syndrome is that of the Typical syndrome.  

6) Therapeutic Options. The sentence “most patients with PanNENs are diagnosed in an advanced state with metastasis” is inaccurate, since it relies on a paper based on a selected population (patients with metastatic and locally advanced disease). That statement can then be misleading. Besides, the entire Section is not specifically related to the topic (functioning PanNENs).

7) Molecular alterations in functioning PanNENs (from page 8, line 373, to page 10, line 440): I wish to sincerely congratulate with the Authors for this extremely usefulk, clear and informative Section.

Author Response

(The authors gave the same response as above.)

Reviewer 4 Report

I would like to congratulate with authors for the efforts spent into drafting a review on Functioning NETs.

The paper is of value. 

There are only some comments

Page 2 lines 49-52 Please re-phrase this part according figure and table 1 about the  role of differentiation into define the grading score.

Authors focus their work on F-NENs. I suggest to remain in this topic above all in the therapeutic part. In particular authors should focus on F-NEN dedicated trials. Please update the reference section on PRRT in F-NENs: https://doi.org/10.3390/cancers14246022

Please add a brief paragraph on the limitations of studies on F-NENs not only from the therapeutic point of view but also into defining the molecular profiles.

Author Response

(The authors gave the same response as above.)

Round 2

Reviewer 2 Report

Specific comments to the authors

In the revised version of the manuscript, the authors could address previous mentioned concerns in a very adequate and convincing manner. Therefore, the revised manuscript “An Insight on Functioning Pancreatic Neuroendocrine Neoplasms" should be accepted, which was a pleasure to review, overall.

Reviewer 3 Report

I really appreciated the profound improvement of the manuscript made by the Authors, and I wish to congratulate with them.

As for the Authors’ Response 1, I have never questioned about considering tumors producing serotonin as functioning. I have simply underlined that, in a classification based on peptide secretion, we should merely recognize that serotonin is not a peptide, no matter the quality of the quoted studies that have employed this classification.

Reviewer 4 Report

Authors addressed all modifications required. This version deserves to be published.